# Diversity of Quill Mites of the Family Syringophilidae (Acariformes: Prostigmata) Parasitizing Starlings of the Genus *Lamprotornis* (Passeriformes: Sturnidae)

**Maciej Skoracki** [1,*], **Milena Patan** [1] , **Markus Unsoeld** [2], **Martin Hromada** [3] , **Zbigniew Kwieciński** [4,5] 
**and Iva Marcisova** [3,*]

1 Department of Animal Morphology, Faculty of Biology, Adam Mickiewicz University, 61-614 Poznań, Poland; milpat@st.amu.edu.pl
2 Zoologische Staatssammlung München, Sektion Ornithologie, Münchhausenstr. 21, 81247 München, Germany; unsoeld@snsb.de
3 Laboratory and Museum of Evolutionary Ecology, Department of Ecology, Faculty of Humanities and Natural Sciences, University of Presov, 08001 Prešov, Slovakia; hromada.martin@gmail.com
4 Department of Ecology and Anthropology, Institute of Biology, University of Szczecin, Wąska 13, 71-415 Szczecin, Poland; zbigniew.kwiecinski@usz.edu.pl
5 Department of Avian Biology and Ecology, Faculty of Biology, Adam Mickiewicz University, Uniwersytetu Poznańskiego 6, 61-614 Poznań, Poland
* Correspondence: maciej.skoracki@amu.edu.pl (M.S.); marcisova.iva@gmail.com (I.M.)

**Abstract:** Quill mites of the family Syringophilidae (Acariformes: Prostigmata) parasitizing starlings of the genus *Lamprotornis* Temminck (Aves: Passeriformes: Sturnidae) from the sub-Saharan region are comprehensively studied for the first time. Among them, two new species are described: (1) *Syringophiloidus soponai* Skoracki, Patan and Unsoeld sp. n., collected from four host species—*Lamprotornis chalybaeus* (Hemprich et Ehrenberg) (type host) in Kenya, Tanzania, and Ethiopia; *L. superbus* (Rüppell) in Kenya and Tanzania; *L. chloropterus* (Swainson) and *L. unicolor* (Shelley) both in Tanzania; (2) *Syringophilopsis parasturni* Skoracki, Patan and Unsoeld sp. n. collected from *L. pulcher* (Müller) and *L. chalcurus* (Nordmann), both in Senegal. Additionally, two *Lamprotornis* species, *L. chalybaeus* in Tanzania and Kenya and *L. chloropterus* in Kenya, are recorded as the new hosts for *Picobia lamprotornis* Klimovicova et al., 2004. We also discussed the diversity of the syringophilid mites associated with starlings.

**Keywords:** acari; birds; diversity; ectoparasites; quill mites; starlings





## 1. Introduction

Quill mites of the family Syringophilidae (Acariformes: Prostigmata) are obligate avian parasites. Without exception, all species reside within the feather quills, where their entire lifecycle, including copulation, takes place. They feed on the fluidic tissue surrounding the feather quill, piercing its wall with their flexible needle-like chelicerae.

Currently, the global fauna of quill mites comprises 443 species grouped into 62 genera [1]. However, despite over half a century of intensive research into the biodiversity of this parasitic group, since Kethley's monograph in 1970 [2], our understanding of these parasites remains as just the tip of the iceberg. Many avian orders have yet to be studied concerning the presence of syringophilid mites [3]. The extent of examination within individual families and genera across various orders is even more limited.

This paper is the first in a planned series of articles offering comprehensive studies on quill mites parasitizing passeriform birds of the family Sturnidae (Passeriformes). Herein, we present our findings on mites associated with birds of the genus *Lamprotornis* Temminck.

The genus *Lamprotornis*, encompassing 22 species of starlings [4], readily distinguished by the vivid and iridescent plumage of its members (Figure 1), is predominantly distributed

across the sub-Saharan region. These birds reveal a wide spectrum of adaptations, enabling them to inhabit diverse ecosystems, from arid savannas to dense forests, thereby reflecting the rich biodiversity and ecological dynamics of their habitats [5]. Unfortunately, the quill mite fauna associated with birds of this genus is still insufficiently explored. To date, only one species of the subfamily Picobiinae, *Picobia lamprotornis* Klimovičová et al., 2014, has been recorded from *Lamprotornis superbus* Rüppell in Kenya [6].

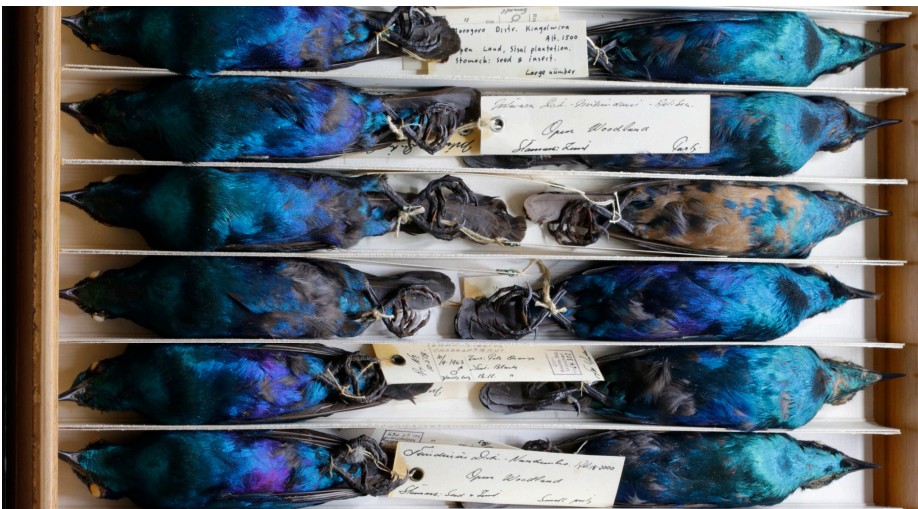

**Figure 1.** Dry bird skins of *Lamprotornis chloropterus*, deposited in the Bavarian State Collection of Zoology, Munich, Germany (ZSM).

## 2. Materials and Methods

### 2.1. Host Sampling

In the present study, we examined all specimens of the *Lamprotornis* genus deposited in the Bavarian State Collection of Zoology, Munich, Germany (SNSB-ZSM) (Figures 1 and 2). The collection comprises the following 16 species:

African Pied Starling *Lamprotornis bicolor* (Gmelin) (*n* = 1).
Ashy Starling *Lamprotornis unicolor* (Shelley) (*n* = 8).
Bronze-tailed Starling *Lamprotornis chalcurus* Nordmann (*n* = 1).
Burchell's Starling *Lamprotornis australis* (Smith, 1836) (*n* = 1).
Cape Starling *Lamprotornis nitens* (Linnaeus) (*n* = 6).
Chestnut-bellied Starling *Lamprotornis pulcher* (Müller) (*n* = 2).
Fischer's Starling *Lamprotornis fischeri* (Reichenow) (*n* = 5).
Golden-breasted Starling *Lamprotornis regius* (Reichenow) (*n* = 3).
Greater Blue-eared Starling *Lamprotornis chalybaeus* Hemprich and Ehrenberg (*n* = 14)
Hildebrandt's Starling *Lamprotornis hildebrandti* (Cabanis) (*n* = 6)
Lesser Blue-eared Starling *Lamprotornis chloropterus* Swainson (*n* = 17).
Long-tailed Glossy Starling *Lamprotornis caudatus* (Müller) (*n* = 2).
Purple Starling *Lamprotornis purpureus* (Müller) (*n* = 3).
Rüppell's Starling *Lamprotornis purpuroptera* Rüppell (*n* = 5).
Splendid Starling *Lamprotornis splendidus* (Vieillot) (*n* = 4).
Superb Starling *Lamprotornis superbus* Rüppell (*n* = 22).

For each avian individual, approximately ten contour feathers proximal to the cloaca, two of upper- or under-tail coverts and a single lesser-wing covert were checked under a stereomicroscope.

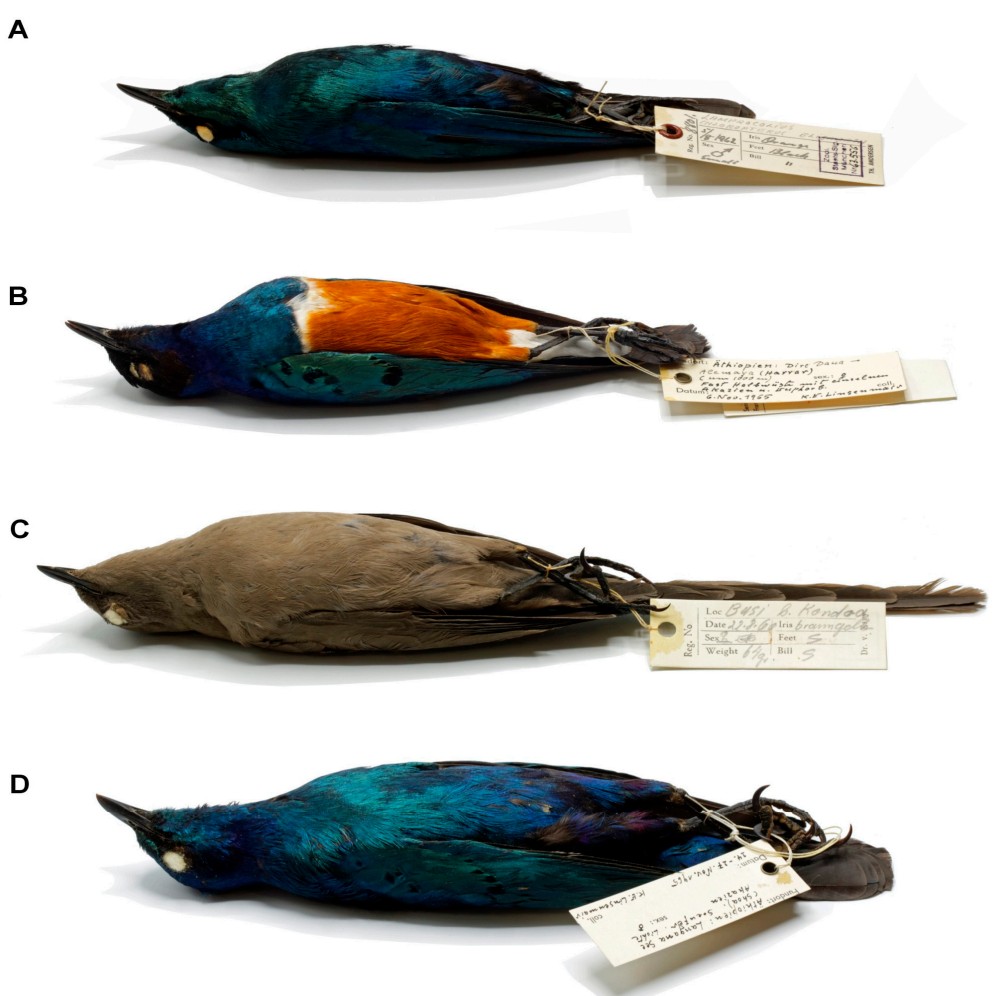

**Figure 2.** Morphological diversity of examined species of the genus *Lamprotornis*. (**A**) *L. chloropterus*; (**B**) *L. superbus*; (**C**) *L. unicolor* and (**D**) *L. chalybaeus*. Studied dry bird skins are deposited in the Bavarian State Collection of Zoology, Munich, Germany (ZSM).

*2.2. Mite Preparation and Morphological Analyses*

Mites found in infested feathers were carefully extracted using fine-pointed tweezers. These specimens were then rendered transparent and pliable by immersion in Nesbitt's solution at ambient temperature for a period of 24 to 36 h [7]. Prior to mounting, mites were briefly transferred to 70% ethanol for approximately 10 min and subsequently embedded on microscopic slides using Hoyer's medium, as per the protocol of Walter and Krantz [8]. Once dried, the slides were sealed with rings and accurately labelled.

The mite specimens underwent examination with a ZEISS Axioscope light microscope outfitted with differential interference contrast (DIC) optics. Illustrations were rendered using a camera lucida attachment. All measurements are presented in micrometers. The dimension ranges of the paratypes are given in parentheses, following the data from the holotype. The idiosomal setation nomenclature aligns with Grandjean's [9] system as modified for Prostigmata by Kethley [10], and the leg chaetotaxy follows Grandjean's [11] classification. All other morphological terminology is in accordance with Skoracki [7].

Descriptive statistics were computed using Quantitative Parasitology on the Web [12], with 95% confidence intervals (the Sterne method).

Mite specimens are curated at the following repositories, abbreviated as: AMU for the Department of Animal Morphology, Adam Mickiewicz University in Poznań, Poland; and SNSB-ZSM for the Section Arthropoda Varia, Bavarian State Collection for Zoology, Munich, Germany.

## 3. Results

*3.1. Descriptions*

3.1.1. *Syringophiloidus saponai* Skoracki, Patan and Unsoeld sp. n.

Female, holotype (Figure 3A,B and Figure 4A–D). Total body length 650 (620–670 in ten paratypes). Gnathosoma. Infracapitulum apunctate. Movable cheliceral digit 140 (140–150) long. Stylophore 180 (180–185) long, exposed portion of stylophore apunctate, 140 (140–145) long. Each medial branch of peritremes has two chambers, each lateral branch has 12 chambers (Figure 4A). Idiosoma. Propodonotal shield well sclerotized with rounded anterior margin and concave posterior margin; surface punctate near bases of setae *ve* and *si*. Propodonotal setae *vi*, *ve* and *si* short and subequal in length. Bases of setae *c1* situated slightly anterior to level of setal bases *se*. Hysteronotal shield well sclerotized and apunctate; anterior margin reaching above level of setal bases *d2*, posterior margin not fused to pygidial shield and not reaching bases of setae *e2*. Bases of setae *d1* situated closer to *d2* than to *e2*. Setae *d2* approximately 1.3 times longer than *e2*. Pygidial shield apunctate and with rounded anterior margin. Genital plate well sclerotized and bearing bases of setae *ag2* and *ag3* on lateral margins. Aggenital setae *ag1* and *ag2* subequal in length, both pairs slightly (1.4 times) shorter than *ag3*. Genital setae *g1* and *g2* equal in length. Pseudanal setae *ps1* and *ps2* equal in length (Figure 4C). Coxal fields I–IV well sclerotized, I–II sparsely punctate or apunctate, III–IV punctate on whole surface. Body cuticular striations as in Figure 3A,B. Legs. Solenidia of legs I as in Figure 4D. Fan-like setae of legs III and IV with seven or eight tines. Lengths of setae: *vi* 30 (25–30), *ve* 30 (30–40), *si* 30 (30–40), *se* 235 (225–265), *c1* 230 (220–245), *c2* 245 (245–285), *d1* 195 (190–205), *d2* 265 (260–285), *e2* 205 (200–215), *f1* 35 (30–35), *h1* 30 (30–35), *h2* (350–415), *ag1* (185–205), *ag2* (185–200), *ag3* 245 (250–280), *ps1* and *ps2* 45 (40–45), *g1* and *g2* 40 (40–45), *3b* 40 (40–50), *3c* (155–160).

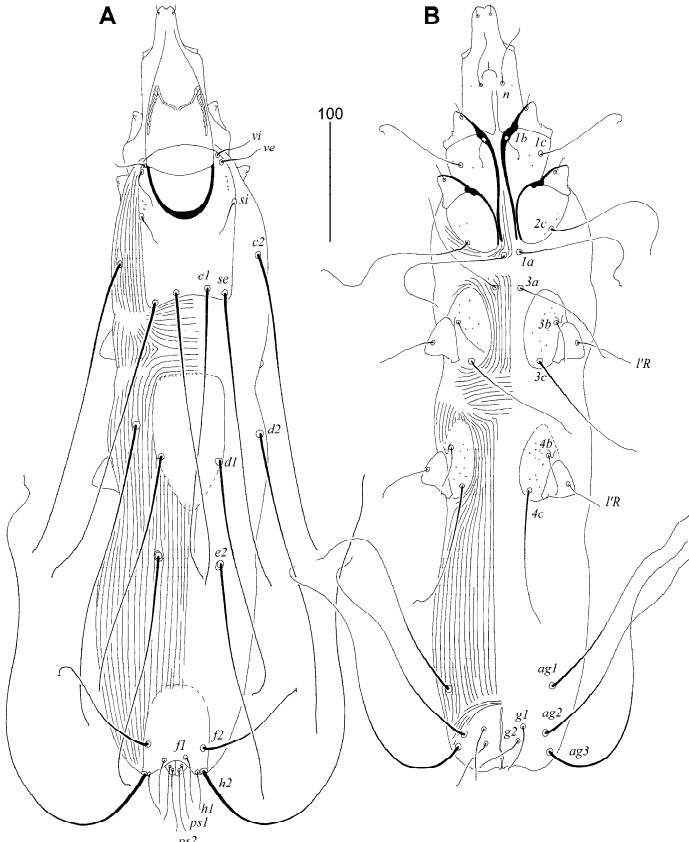

**Figure 3.** *Syringophiloidus saponai* Skoracki, Patan and Unsoeld sp. n., female. (**A**) dorsal view; (**B**) ventral view.

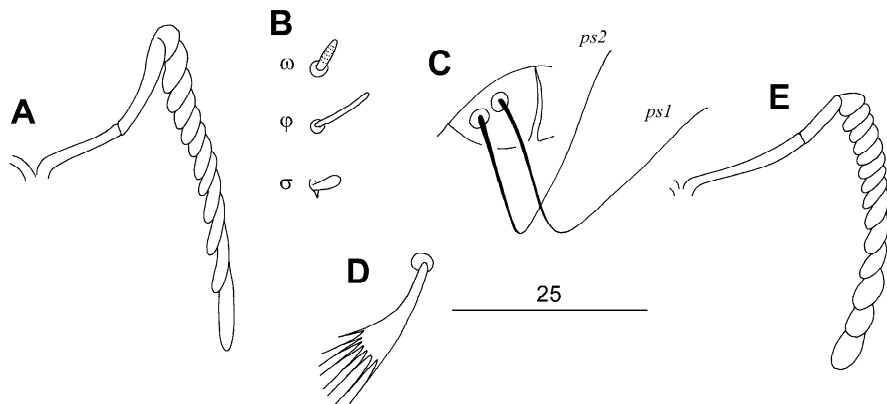

**Figure 4.** *Syringophiloidus saponai* Skoracki, Patan and Unsoeld sp. n. female (**A–D**): (**A**) peritreme; (**B**) solenidia of leg I; (**C**) pseudanal setae; (**D**) fan-like seta *p'III*. male (**E**): (**E**) peritreme.

Male (Figures 4E and 5A,B). Total body length 490–510 in two paratypes. Gnathosoma. Infracapitulum apunctate. Movable cheliceral digit 130 long. Stylophore 150–155 long; exposed portion of stylophore apunctate, 120–125 long. Each medial branch of peritremes has two chambers, each lateral branch has 14 chambers (Figure 4E). Idiosoma. All dorsal shields and coxal fields apunctate. All idiosomal setae smooth. Propodonotal shield well sclerotized and rectangular. Propodonotal setae *vi*, *ve* and *si* subequal in length. Bases of setae *c1* and *se* situated on same transverse level. Hysteronotal shield well sclerotized, large, not fused to pygidial shield; anterior margin reaching above level of setal bases *d2*, posterior margin reaching level of setal bases *e2*. Bases of setae *d1* situated equidistant between bases of setae *d2* and *e2*. Setae *d2* about 11 times longer than *d1* and *e2*. Pygidial shield rounded anteriorly. Setae *h2* about 10–12 times longer than *f2*. Length ratio of setae *ag1*:*ag2*:*ag3*, 3:1:2. Body cuticular striations as in Figure 5A,B. Lengths of setae: *vi* 60–70, *ve* 60–70, *si* 50–65, *se* 280, *c1* 260, *c2* 230–255, *d1* 25, *d2* 270, *e2* 25, *f2* 20–25, *h2* 240–260, *ag1* 125–150, *ag2* 50–55, *ag3* 70–75, *3b* 40–45 and *3c* 135.

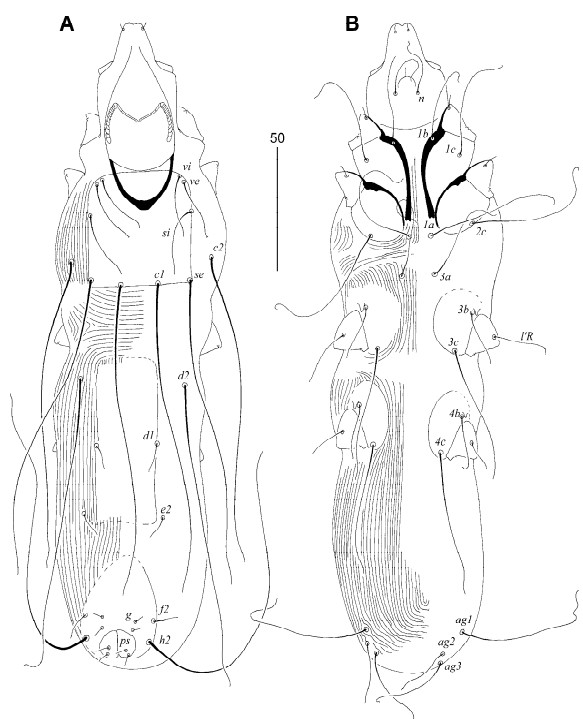

**Figure 5.** *Syringophiloidus saponai* Skoracki, Patan and Unsoeld sp. n., male. (**A**) dorsal view; (**B**) ventral view.

Type Material

Female holotype, 12 female paratypes and two male paratypes (reg. no. MS 22-0821-004) from wing covert quill of the Greater Blue-eared Glossy-Starling *Lamprotornis chalybaeus* (Hemprich and Ehrenberg) (host reg. no. SNSB-ZSM 26.404; male); KENYA: Western Region, Mount Elgon National Park, 2300 m a.s.l., 8 February 1925, coll. S. Alinder.

Type Material Deposition

Holotype and paratypes are deposited in the SNSB-ZSM, except for four female paratypes and one male paratype in the AMU.

Additional Material

Eight females and two males (reg. no. MS 22-0821-002) from wing covert quill of type host species (host reg. no. SNSB-ZSM 2751; female); TANZANIA: Arusha Region, Arusha District, Sonja, 6 March 1911, coll. Kattwinkel. Ten females (reg. no. MS 22-0821-003) from the same habitat and host species (host reg. no. SNSB-ZSM 66.17; male); ETHIOPIA: Oromia Region, Jimma District, Jimma, 21 November 1965, coll. K. E. Linsenmair.

Two females (reg. no. MS 21-1012-049) from under-tail covert quill of the Superb Starling *Lamprotornis superbus* (Rüppell) (host. reg. no. SNSB-ZSM 2737); TANZANIA: Mara Region, Serengeti District, near Ikoma, February 1911, coll. Kattwinkel. Seven females and four males (reg. no. MS 23-0925-001) from wing covert quill of the same host species (field reg. no. BB15129); KENYA: Rift Valley Province, Nakuru County, northwest of Gilgil, Soysambu Conservancy (FSC), 13 May 2014, coll. W. Wamiti.

Five females (reg. no. MS 22-0821-011) from wing covert quill of the Lesser Blue-eared Glossy-Starling *Lamprotornis chloropterus* (Swainson) (host reg. no. SNSB-ZSM 60.570; male); TANZANIA: Morogoro Region, Morogoro District, Kingolwira, 30 September 1959, alt. 1500 ft, coll. Th. Andersen. One female (reg. no. MS 23-1120-005) from the same host species (host reg. no. SNSB-ZSM 66.575); TANZANIA: Mtwara Region, Mtwara District, Mikindani, 6 April 1966, coll. Th. Andersen. Two females (reg. no. MS 23-1120-006) from the same host species (host reg. no. SNSB-ZSM 64.302); TANZANIA: Ruvuma Region, Tanduru District, Nandembo, alt 610 m. a.s.l., 10 September 1963, collected by Th. Andersen.

Five females and one male (reg. no. MS 21-1012-050) from quill of under-tail covert of the Ashy Starling *Lamprotornis unicolor* (Shelley) (host reg. no. SNSB-ZSM 60.569; male); TANZANIA: illegible location on the label, alt. 945 m a.s.l., 19 June 1959, coll. Th. Andersen.

Differential Diagnosis

*Syringophiloidus saponai* sp. n. belongs to the "*glandarii*-species-group" [7] by possessing two–three elongated chambers in each medial branch of the peritremes. Among the species of this group, *S. saponai* is morphologically the most similar to *S. dendrocittae* Fain, Bochkov and Mironov, 2000, described from *Dendrocitta vagabunda* (Passeriformes: Corvidae) from E. Asia [13], in having short setae *vi*, *ve* and *si* (less than 50 μm). The new species differs from *S. dendrocittae* by the following features: in females of *S. saponai*, each lateral branch of the peritremes has 12 chambers, the lengths of hysteronotal setae *d1*, *d2* and *e2* are 190–205, 260–285 and 200–215, respectively; in males, hysteronotal setae *d2* are about 11 times longer than *d1* and *e2*. In females of *S. dendrocittae*, each lateral branch of the peritremes has nine chambers, the lengths of hysteronotal setae *d1*, *d2* and *e2* are 94, 157 and 132, respectively; in males, hysteronotal setae *d1*, *d2* and *e2* are short and subequal in the length. *Syringophiloidus saponai* can be easily distinguished from the other species associated with the starlings, i.e., *S. presentalis* Chirov and Kravtsova, 1995, by the length ratio of setae *vi:si* 1:1 (versus 1:6.5 in *S. presentalis*).

Etymology

The species name "soponai" is derived from the deity "Sopona"—the African god of smallpox and skin diseases.

### 3.1.2. *Syringophilopsis parasturni* Skoracki, Patan and Unsoeld sp. n.

Female, holotype (Figure 6). Total body length 1065 (1100 in one paratype). Gnathosoma. Infracapitulum apunctate. Hypostomal apex ornamented by pair small and sharp-ended protuberances (Figure 6C). Movable cheliceral digit 170 (175) long, harpoon-like, each with three teeth (Figure 6D). Stylophore 250 (260) long; exposed portion of stylophore apunctate, 220 (230) long. Each medial branch of peritremes has three chambers, each lateral branch has ten chambers. Idiosoma. Propodonotal shield well sclerotized and punctate laterally. Bases of setae *c2* situated posterior to level of setae *si*; bases of setae *c1* and *se* situated on same transverse level. Length ratio of setae *vi:ve:si*, 1:1.2–1.4:3.7–3.9. Hysteronotal shield reduced to two small and apunctate sclerites surrounding bases of setae *d1*. Bases of setae *d1* situated closer to *e2* than to *d2*. Setae *d1*, *d2* and *e2* subequal in length. Pygidial shield sparsely punctate near setal bases *h1* and *h2* or apunctate; anterior margin indistinct. All aggenital setae (*ag1–3*) long (longer than 180), setae *ag1* slightly (1.3 times) longer than *ag2*. Genital setae *g1* and *g2* are equal in length. Pseudanal setae *ps1* and *ps2* equal in length. Coxal fields I–IV well sclerotized and apunctate. Body cuticular striations as in Figure 6A,B. Legs. Solenidia of legs I as in Figure 6E. Fan-like setae of legs III and IV with 11–12 tines. Lengths of setae: *vi* 100 (100), *ve* 145 (120), *si* 390 (370), *se* 380 (370), *c1* 400, *c2* > 330, *d1* 385, *d2* 420, *e2* 330, *f1* 40 (45), *h1* 75 (80), *f2* 405, *h2* > 340, *ag1* 280, *ag2* 185, *ag3* > 260, *ps1* and *ps2* 45 (40), *g1* and *g2* (60) and *l'RIII* 120 (90).

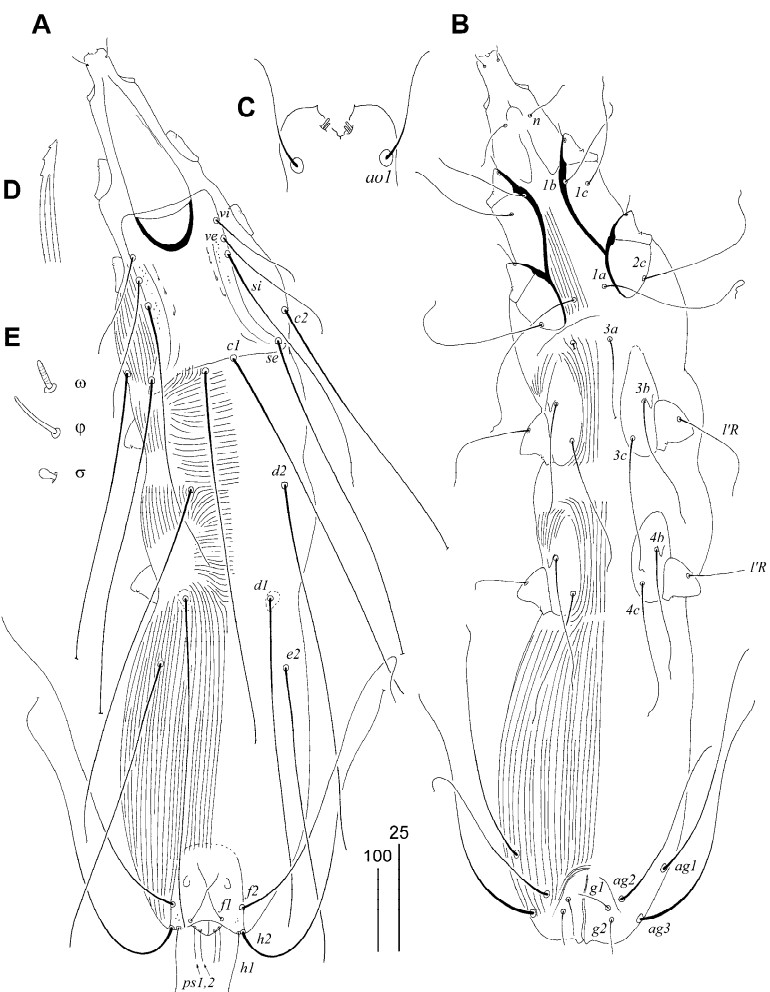

**Figure 6.** *Syringophilopsis parasturni* Skoracki, Patan and Unsoeld sp. n., female. (**A**) dorsal view; (**B**) ventral view; (**C**) hypostomal apex; (**D**) distal tip of movable cheliceral digit; (**E**) solenidia of legs I.

Male. Not found.

Type Material

Female holotype and one female paratype (reg. no. MS 21-1012-048) from wing-covert quill of the Chestnut-bellied Starling *Lamprotornis pulcher* (Müller) (host at ZSM uncatalogued); SENEGAL: no other data.

Type Material Deposition

Female holotype is deposited in the SNSB-ZSM and female paratype in the AMU.

Additional Material

One female (reg. no. MS 22-0821-006) from wing covert quill of the Bronze-tailed Glossy-Starling *Lamprotornis chalcurus* (Nordmann) (host at ZSM uncatalogued); SENEGAL: no other data.

Differential Diagnosis

This new species belons to the "*turdi*-species-group", of which females possess setae *h2* and *f2* distinctly longer than *f1* and *h1* [14]. *Syringophilopsis parasturni* sp. n. is morphologically the most similar to *S. sturni* Chirov and Kravtsova, 1995, described from *Sturnus vulgaris* Linnaeus (Passeriformes: Sturnidae) [15] and known from the Palaearctic localities [7]. In females of both species, the infracapitulum is apunctate, the hypostomal apex is ornamented by a pair of small and sharp-ended protuberances and two small hysteronotal sclerites surround bases of setae *d2* are present. The new species differs from *S. sturni* by the following features: in females of *S. parasturni*, each lateral branch of the peritremes has ten chambers, the propodonotal setae *ve* are 120–145 μm long, the bases of setae *c1* and *se* are situated on the same transverse level and the coxal fields I–IV are apunctate. In females of *S. sturni*, each lateral branch of the peritremes has 13–14 chambers, bases of setae *se* are situated distinctly anterior to the level of setal bases *c1* and the coxal fields I–IV are punctate.

Etymology

The name "parasturni" is the combination of the prefix "para" (meaning similar, close to) and "sturni" that together indicate the similarity of the new species with *Syringophilopsis sturni* described by Chirov and Kravtsova [15].

3.1.3. *Picobia lamprotornis* Klimovičová, Skoracki, Wamiti and Hromada, 2014

This species was described based on the material of both sexes collected from *Lamprotornis superbus* in Kenya [6], and to this point in time, there were no other records of this mite since the first description. Below, we report two new host species (*L. chalybaeus* and *L. chloropterus*) and a new locality (Tanzania) for this quill mite.

Material Examined

One female (non-physogastric form), two females (physogastric form) and one male (reg. no. MS 22-0821-001) from the contour feather quill of the Greater Blue-eared Glossy-Starling *Lamprotornis chalybaeus* (Hemprich and Ehrenberg) (host reg. no. ZSM 2751; female); TANZANIA: Arusha Region, Arusha District, Sonja, 6 March 1911, coll. Kattwinkel. One female (physogastric form) (reg. no. MS 22-0821-005) from the same habitat and host species (host reg. no. ZSM 26.405; female); KENYA: Western Region, Mount Elgon National Park, alt. 2300 m. a.s.l., 5 February 1925, coll. S. Alinder.

One female (physogastric form) and one male (reg. no. MS 22-0821-010) from contour feather quill of the Lesser Blue-eared Glossy-Starling *Lamprotornis chloropterus* (Swainson) (host reg. no. ZSM 66.575; female); TANZANIA: Mtwara Region, Mtwara District, Mikindani, 6 April 1966, coll. Th. Andersen. Two females (physogastric form) (reg. no. MS 22-0821-009) from the same habitat and host species (host reg. no. ZSM 64.302; male);

TANZANIA: Ruvuma Region, Tunduru District, Nandembo, alt 610 m. a.s.l., 10 September 1963, coll. Th. Andersen. One female (physogastric form) (reg. no. MS 22-0821-008) from the same habitat and host species (host at ZSM uncatalogued; male); TANZANIA: Morogoro Region, Morogoro District, 8 July 1955, alt. 305 m. a.s.l., coll. Th. Andersen. Two females (non-physogastric form) and three females (physogastric form) (reg. no. MS 22-0821-007) from the same habitat and host species (host reg. no. ZSM 64.303; male); TANZANIA: Ruvuma Region, Tunduru District, Nandembo, alt. 610 m. a.s.l., 10 September 1963, coll. Th. Andersen.

Two females (non-physogastric form) (MS 23-1120-001) from the Superb Starling *Lamprotornis superbus* (field no. BB13607); KENYA: Rift Valley Region, Laikipia District, Laikipia Nature Conservancy (LWEC), 1 July 2013, coll. W. Wamiti. One female (physogastric form) (reg. no. MS 23-1120-002) from the same host species (field no. BB13669); KENYA: Rift Valley Region, Laikipia District, Ol Pejeta Conservancy (Golf 8), 13 April 2014, coll. W. Wamiti. One female (physogastric form) (reg. no. MS 23-1120-003) from the same host species (field no. BB13664) and other data. One female (physogastric form) and one female (non-physogastric form) (reg. no. MS 23-1120-004) from the same host species (field no. BB13599); KENYA: Rift Valley Region, Laikipia District, Laikipia Nature Conservancy (Centre), 29 June 2013, coll. W. Wamiti.

### 3.2. Prevalence

In our study, we examined one hundred individual hosts encompassing sixteen species within the genus *Lamprotornis* (73% of total number of species), finding that six of these species (representing 37%) were hosts of quill mites. The prevalence rates of infestation are presented in Table 1. The following host species were not infested by quill mites: *L. bicolor*, *L. australis*, *L. nitens*, *L. fischeri*, *L. regius*, *L. hildebrandti*, *L. caudatus*, *L. purpureus*, *L. purpuroptera* and *L. splendidus*.

**Table 1.** Starling birds of the genus *Lamprotornis* and their quill mite parasites with index of prevalence, confidence interval (CI$^{\text{Sterne method}}$), and occupying habitat.

| Host Species, No. of Examined Specimens | No. of Infested Specimens | Prevalence (CI) | Habitat | Quill Mite Species |
|---|---|---|---|---|
| *L.chalybaeus*, n = 14 | 2 | 14.3% (3.0–41.4) | cov | *S. saponai* sp. n. |
| " | 1 | 7.1% (0.1–31.7) | con | *P. lamprotornis* |
| " | 1 | 7.1% (0.1–31.7) | cov + con | *S. saponai* sp. n. + *P. lamprotornis* |
| Total | 4 | 28.6% (10.4–57.4) | con, cov | *S. saponai* sp. n., *P. lamprotornis* |
| *L. superbus*, n = 22 | 2 | 9.1% (1.6–29.1) | cov | *S. saponai* sp. n. |
| " | 3 | 13.6% (3.8–33.8) | con | *P. lamprotornis* |
| Total | 5 | 22.7% (9.4–45.3) | con, cov | *S. saponai* sp. n., *P. lamprotornis* |
| *L. chloropterus*, n = 17 | 1 | 5.9% (0.3–28.7) | cov | *S. saponai* sp. n. |
| " | 2 | 11.8% (2.1–35.0) | con | *P. lamprotornis* |
| " | 2 | 11.8% (2.1–35.0) | con + cov | *S. saponai* sp. n. + *P. lamprotornis* |
| Total | 5 | 29.4% (12.4–54.4) | con, cov | *S. saponai* sp. n., *P. lamprotornis* |
| *L. unicolor*, n = 8 | 1 | 12.5% (0.6–50.0) | con | *S. saponai* sp. n. |
| *L. pulcher*, n = 2 | 1 | 50% (2.5–97.5) | cov | *S. parasturni* sp. n. |
| *L. chalcurus*, n = 1 | 1 | 100% (50–100) | cov | *S. parasturni* sp. n. |

cov = wing-covert quills; con—contour quills.

## 4. Discussion

*Species richness of quill mites associated with starlings.* The starlings currently serve as hosts for nine species of quill mites belonging to two subfamilies: Picobiinae Johnston and Kethley, 1973 and Syringophilinae Lavoipierre, 1953 (Table 2).

**Table 2.** Quill mites of the family Syringophilidae associated with birds of the family Sturnidae.

| Quill Mite Species | Host Species | Locality | References |
|---|---|---|---|
| **Subfamily Picobiinae Johnston and Kethley, 1973** | | | |
| *Picobia indonesiana* Skoracki and Glowska, 2008 | *Aplonis panayensis* (Scopoli) * | Orie: Indonesia | [16] |
| " | *Enodes erythrophris* (Temminck) | Orie: Indonesia (Sulawesi) | [16] |
| " | *Mino dumontii* (Lesson) | Ocea: Indonesia (New Guinea) | [16] |
| *Picobia sturni* Skoracki, Bochkov and Wauthy, 2004 | *Sturnus vulgaris* (Linnaeus) * | Pala: Poland, Slovakia, Moldova | [7,17] |
| " | *Sturnus unicolor* (Temminck) | Pala: Morocco | [18] |
| " | *Spodiopsar cineraceus* (Temminck) | Pala: Japan | [7] |
| *Picobia lamprotornis* Klimovičová, Skoracki, Wamiti and Hromada, 2014 | *Lamprotornis superbus* (Rüppell) * | Afro: Kenya | [6], c.s. |
| " | *Lamprotornis chalybaeus* (Hemprich & Ehrenberg) | Afro: Tanzania, Kenya | c.s. |
| " | *Lamprotornis chloropterus* (Swainson) | Afro: Tanzania | c.s. |
| **Subfamily Syringophilinae Lavoipierre, 1953** | | | |
| *Syringophiloidus graculae* Fain, Bochkov and Mironov, 2000 | *Gracula religiosa* (Linnaeus) | Orie: SE Asia | [13] |
| *Syringophiloidus presentalis* Chirov and Kravtsova, 1995 | *Sturnus vulgaris* (Linnaeus) | Pala: France, Poland, Slovakia, Kyrgyzstan; Near: USA | [7,15,19] |
| *Syringophiloidus saponai* sp. n. | *Lamprotornis chalybaeus* (Hemprich & Ehrenberg) * | Afro: Kenya, Tanzania, Ethiopia | c.s. |
| " | *Lamprotornis superbus* (Rüppell) | Afro: Tanzania, Kenya | c.s. |
| " | *Lamprotornis chloropterus* (Swainson) | Afro: Tanzania | c.s. |
| " | *Lamprotornis unicolor* (Shelley) | Afro: Tanzania | c.s. |
| *Syringophilopsis sturni* Chirov and Kravtsova, 1995 | *Sturnus vulgaris* (Linnaeus) | Pala: Poland, Kazakhstan, Kyrgyzstan, Ukraine | [7,15] |
| *Syringophilopsis parasturni* sp. n. | *Lamprotornis pulcher* (Müller) * | Afro: Senegal | c.s. |
| " | *Lamprotornis chalcurus* (Nordmann) | Afro: Senegal | c.s. |
| *Krantziaulonastus buczekae* (Skoracki, 2002) | *Sturnus vulgaris* (Linnaeus) | Pala: Poland | [20] |

"*"—type host species; c.s.—current study.

Quill mites of the subfamily Picobiinae associated with the starlings are represented by three species of the genus *Picobia*, which is widely distributed on passeriform birds inhabiting the quills of contour feathers. Among these, *Picobia lamprotornis* is an oligoxenous parasite, infesting multiple hosts but confined to the genus *Lamprotornis* ([6], current study). The remaining two *Picobia* species are mesostenoxenous, infesting multiple host genera but restricted to the family Sturnidae. *Picobia indonesiana* is associated with birds of the genera *Aplonis*, *Enodes* and *Mino*, whereas *Picobia sturni* is linked to the genera *Sturnus* and *Spodiopsar* [7,16]. Regarding the distribution of Picobiinae parasites associated with starlings, each species occupies a distinct zoogeographical region. *Picobia indonesiana* is found with Indonesian hosts (Oriental/Oceanian region), and *P. sturni* is a Palearctic species. However, the presence of *P. sturni* on hosts like *Sturnus vulgaris*, originally native to Eurasia but now cosmopolitan and invasive, is plausible. This host species, the common starling, is found on every continent except Antarctica and in regions like North America, Argentina, South Africa, Eastern Australia and New Zealand [21,22]. The third picobiine species, *Picobia lamprotornis*, is a sub-Saharan endemic, associated with African birds of the genus *Lamprotornis*. Whether this species also infests other African starlings from genera such as

*Onychognathus*, *Poeoptera* and *Hylopsar* remains as an unanswered question [4,5]. Notably, apart from *Lamprotornis*, which was the focus of this research, none of the 13 genera native to sub-Saharan Africa have been examined for mite infestation [1].

Mites of the subfamily Syringophilinae associated with starlings comprise six species from three genera: *Syringophiloidus* Kethley, 1970, *Syringophilopsis* Kethley, 1970 and *Krantziaulonastus* Skoracki, 2011. To date, all these species have been documented as monoxenous parasites, each recorded from a single host species. However, this may be the result of insufficient study of closely related host species. For instance, *Syringophiloidus graculae* has been identified solely from *Gracula religiosa* [13], but other host species within the *Gracula* genus have not yet been examined [1]. A similar situation may apply to *Syringophiloidus presentalis*, *Syringophilopsis sturni* and *Krantziaulonastus buczekae;* each is only associated with *Sturnus vulgaris*. It remains unconfirmed whether these species are also present on *Sturnus unicolor*. Both new species described in this article, *Syringophiloidus saponai* and *Syringophilopsis parasturni*, should be considered, akin to *Picobia lamprotornis*, as oligoxenous sub-Saharan endemics associated with birds of the genus *Lamprotornis*.

*Habitat specificity and simultaneous infestations of syringophilid mites.* Syringophilid mites demonstrate a remarkable evolutionary adaptation in their choice of habitat within the plumage. These mites are found in the quills of various feathers, including primaries, secondaries, alulars, coverts, rectrices and body feathers [7,23,24]. Intriguingly, the subfamilies Syringophilinae and Picobiinae exhibit preferences for distinct types of feathers. Picobiines exclusively occupy body feathers, while members of the syringophilines are more versatile, residing in the quill feathers of both wings and tail. This preference likely reflects an evolutionary divergence in the early stages of syringophilid evolution [25–27]. Individual birds (especially passerines) may host multiple syringophilid species from two to four genera, yet each mite species is highly specialized, inhabiting quills of a specific feather type, distinct from those occupied by others [2,28–30]. This niche specialization is thought to be determined primarily by two key feather characteristics: the volume of the quill and the thickness of its wall [23,31]. For instance, smaller mites with short chelicerae cannot pierce the walls of very large quills that leads to their inevitable demise. Conversely, the limitation on quill volume can constrain the size of mite populations. Obviously, there is a correlation between quill volume and wall thickness, and between mite size and chelicerae length. The optimal niche allows a relatively large number of mites to be housed while ensuring the quill wall thickness does not hinder feeding [23,24,31].

A compelling example of multi-infestation is the Eurasian Starling *Sturnus vulgaris*, parasitized by four quill mite species (Table 2), each inhabiting a different feather type: *Syringophilipsis sturni* (total body length 1035–1190 µm) in primaries and large secondaries, *Syringophiloidus presentalis* (800–820 µm) in smaller secondaries and greater wing coverts, *Krantziaulonastus buczekae* (550–580 µm) in small wing- and undertail coverts and *Picobia sturni* (520–645 µm) as a member of subfamily Picobiinae, always in quills of body feathers [7,17]. In our investigation of *Lamprotornis* birds, we encountered multi-infestations of quill mites on *L. chalybaeus*, *L. chloropterus*, and *L. superbus*. This case of multi-infestation involved two mite species: *Picobia lamprotornis*, which inhabits the quills of body feathers, and *Syringophiloidus saponai*, found inside the quills of wing coverts (see Table 1). Each species infested its typical habitat in the bird plumage. A surprising finding was the detection of a representative of the genus *Syringophilopsis* in the wing coverts. These large mites typically do not infest such feathers, as they are too small to support a sufficiently large progeny. Our observation likely represents a random colonization of the coverts by foundress females of the infrapopulation, resulting in a low mite count in this marginal habitat. This phenomenon is sporadically observed (MS—unpublished observation) in cases of high infestation intensity, where all feathers of the type habitat are already occupied by mites. However, we could not confirm this, as we did not collect flight feathers from the wings, the typical habitat for *Syringophilopsis* species. Also, the limited number of available and examined host individuals (*Lamprotornis pulcher* (*n* = 2) and *L. chalcurus* (*n* = 1)) does not allow for far-reaching conclusions.

This preliminary analysis highlights the complexity of quill mite infestations in avian hosts, marked by habitat specificity and co-infestations. The findings suggest a sophisticated ecological interplay, where different mite species exploit distinct niches within the same host, minimizing direct competition. This study underscores the importance of considering habitat specificity and ecological interactions in understanding the dynamics of parasite–host relationships in avian species. Further research, particularly including a broader range of feather types, could provide deeper insights into these complex ecological patterns.

*Prevalence*. This updated analysis assesses the prevalence of quill mite infestations in various bird host species, specifically *L. chalybaeus*, *L. superbus* and *L. chloropterus*, excluding species with very small sample sizes to ensure more reliable results. *Lamprotornis chalybaeus* exhibits a total prevalence of 28.6%, which indicates a moderate to high level of infestation within this species. *Lamprotornis superbus* shows a total prevalence of 22.7% which suggests a slightly lower prevalence of infestation compared to *L. chalybaeus*, though still significant. *Lamprotornis chloropterus* has a total prevalence of 29.4%, and this prevalence is comparable to that of *L. chalybaeus*, indicating a similar level of infestation susceptibility. The prevalence rates across these host species are notably similar, and might suggest a consistent level of vulnerability to quill mite infestations across these bird species, possibly due to similar ecological niches or environmental conditions that these mite species share. Obtained results show a rather high index of prevalence compared with other passerine hosts, e.g., from the families Nectariniidae, Estrildidae and Corvidae [32–35]. The relatively high prevalence of quill mite infestations can be partially attributed to the social nature of these host species. For example, *L. chalybaeus* is known to form pairs and small flocks, occasionally congregating in groups of up to 300 individuals at fruiting trees and forming large roosts post-breeding, exceeding 400 individuals [36]. *Lamprotornis chloropterus* exhibits even more pronounced social behavior, forming roosts of 500–1200 individuals in the non-breeding season, with reports of roosting flocks exceeding 10,000 in Zambia [37]. The formation of large flocks and roosts provides an ideal environment for the transmission of parasites, including quill mites, due to increased contact rates among individuals. However, it should be taken into account that the frequency results obtained from these studies may be lower than the actual values. This is primarily because syringophilid mites do not infest every host in nature nor do they inhabit all feathers in the plumage of their hosts. To accurately determine index of prevalence, it is necessary to examine a wide range of bird individuals, taking into account factors such as age, season and location, and to analyze as many feathers as possible [38–40]. However, feather samples taken from both ornithological collections and live birds inherently allow for the examination of only a limited selection of feathers. Therefore, to alleviate this limitation, further research on the habitat specificity of mites is crucial.

**Author Contributions:** Conceptualization, M.S., I.M. and M.H.; methodology, M.S., M.P. and M.U.; investigation, M.S., I.M., M.P., M.U., Z.K. and M.H.; resources and material collection, M.S., M.P. and M.H.; visualization, M.S. and M.U.; writing—original draft preparation, M.S., I.M., M.P., M.U., Z.K. and M.H.; writing—review and editing, M.S., I.M., M.P., M.U., Z.K. and M.H.; supervision, M.S., I.M. and M.H.; project administration and funding acquisition, M.H. All authors have read and agreed to the published version of the manuscript.

**Funding:** Slovak Research and Development Agency under the contract APVV-22-0440 and the Agency of the Ministry of Education, Research and Sport of the Slovak Republic and Slovak Academy of Sciences 1/0876/21.

**Institutional Review Board Statement:** Ethical review and approval were waived for this study due to the use of only dead animals (specimens deposited in the ornithological collection).

**Data Availability Statement:** All necessary data (such as localities) are available in the text of this article.

**Acknowledgments:** We express our gratitude to all members of the Society—Freunde der Zoologis-chen Staatssammlung München e. V.—for their invaluable support during our research tenure at the Bavarian State Collection of Zoology, Munich, Germany. We also thank the anonymous reviewers for their critical review of the manuscript.

**Conflicts of Interest:** The authors declare no conflict of interest.

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
