# Peer review of "Diversity of Quill Mites of the Family Syringophilidae (Acariformes: Prostigmata) Parasitizing Starlings of the Genus Lamprotornis (Passeriformes: Sturnidae)"

_diversity, doi:10.3390/d16010051_

Round 1
Reviewer 1 Report
Comments and Suggestions for Authors
This is a nice study, also very well written. A few minor comments. Follow them if you agree, or feel free to reject them of you think I am wrong.
Line 62: delete the word 'avian'
Lines 66-84: The meaning of '(N = 1)' is self-evident, nedd not be explained. But what does the following number (e.g. '0' in line 66) mean? Please explain.
Line 89: What doeas the word 'sample' mean? Is it an individual? If yes, please correct to 'For each bird individual...'
Lines 89-91: To what extent such studies are destructive or non-destructive to museum skins? Please tell the readers whether these feathers are sacrified (destroyed) for the purpose of the study.
Line 101: the appears to be sentence fragment.
Line 326: 'birds of the family Sturnidae' can be replaced by 'Sturnid birds'
Line 363: 'distinct plumage preferences' replace by 'preferences for distinct types of feathers'
Line 377: 'multiinfestation' change to 'multiple infestation'
Lines 430-433: Of course, you are right, incrasing sample sizes would make confidence intervals narrower (give more accurate estimation). But there is another point worth to mention. Since you can study only a very small number of feathers, these results probably underestimate the true prevalence. ('False negative diagnosis' probably occurs)
Comments on the Quality of English Language
Very well written.
See my few minor suggestions above.
Author Response
Dear Reviewer,
Thank you very much for your time and all the comments and corrections to our manuscript. We have considered all the remarks and have incorporated the corrected text into the new version of the manuscript.
Once again, thank you very much.
Sincerely,
Maciej Skoracki
Reviewer 2 Report
Comments and Suggestions for Authors
The reviewed manuscript provide the analysis of diversity, host associations, distribution and to some extent the pervalence of quill mites of the family Syringophilidae associated with starlings of the genus Lamprotornis distributed in Africa. Additionally, two new syringophiline species from the genera Syringophiloidus and Syringophilopsis are described. The manuscript presents interesting and useful results, generally well organized and written. However, some parts need moderate linguistic corrections, and descriptions of the new species are written a bit uncommonly from the taxonomic standards view. The manuscript can be accepted after a minor but careful revision. Main comments are given below, other recommendations and corrections are directly in the text file (attached).
P. 2. Material and Methods, lines 65-81, list of examined Lamprotornis species and number of examined and infected individuals.
It would be more reasonable to give this list in a table.
P. 2. (Line 83). Caption to Figure 1: Dry bird skins of birds of the genus Lamprotornis …
The figure actually shows only one host species. The caption should be edited.
P. 4-9. (Starting from Line 115 and though all subsequent text up to page 9). “Syringophiloidus saponai Skoracki, Patan and Unsoeld sp. n. ….”
A)In the previous text of the manuscript (pages 1-3), all Latin names of bird and mite species and genera are given, as it is always used in zoological issues, in Italics. Starting from the line 115 and up to the end of page 9, all Latin names of species and genera are unexpectedly appear in the regular font. It should be corrected throughout the text. If it is not error or carelessness of the authors, and they just followed the style of the journal “Diversity”, the style of the journal is quite uncommon, because it does not correspond to common standards of taxonomic zoological papers with descriptions.
B)The same can be noted regarding the setal marks. According to common standards of descriptions for many groups of arthropods including Acari, setal marks are usually shown in Italics that helps better recognizing them in the text. However, throughout all descriptions setal marks are in the regular font. Is it error of author or the style of the journal.
F)Finally, descriptions of both new species are given as a plain text, according to all rules of English. This is also in a strong contrast with standards of zoological descriptions, which should be made in a telegraph style (i.e., minimum of verbs, the verb “to be” is not used, articles in most cases are omitted). If these descriptions are made according to the style of the journal, the descriptions look really odd.
P. 7, line 205. “…by the presence short setae vi, ve and si (less than 50 μm).”
This text should be edited like “... in having setae vi, ve and si short (less than 50 μm).” The character state here is the lengths of setae, but not their presence. They are always present in these mites.
P. 9, line 269, 270. “The name "parasturni" is taken from the morphologically closely related species Syringophilopsis sturni described by Chirov and Kravtsova [15].”
It incorrect statement. The epithet "sturni" in S. sturni is really derived from the name of its host species. While in the new species, the specific epithet is the combination of the prefix "para" (meaning similar, close to) and "sturni" that altogether indicates the similarity of the new species with S. sturni. So, the new name is not "taken".
P. 11, line 337, 338. “However, its presence on hosts like Sturnus vulgaris, originally native to Eurasia but now cosmopolitan 338 and invasive, is plausible.”
From this and previous phrase, it is not clear what mite species do you mean here, P. sturni or P. indonesiana? Give exact name of mite in this sentence.
P. 11, line 339, 340. “This host species is found on every continent except Antarctica and in regions like North America, Argentina, South Africa, eastern Australia, and New 0 Zealand.”
As above, the name of host is unclear from the previous context. It is better to specify exactly the name (English or Latin) of the host you mean here.
P. 12, lines 373, 374. “Interestingly, there is a correlation between quill volume and wall thickness, and between mite size and 374 chelicerae length.”
The first correlation pair is not suprizing. Than larger bird, then larger feather, and then thicker the walls of quill.
Figures:
In Figs. 3, 5, 6 – it would be reasonable and useful to figure legs at least on one side of the mite body.
In Fig. 4, setal designation “n” on subcapitulum is missed.
Fig. 6. Captions for fragments designated C, D, E are lost.

Comments on the Quality of English LanguageThe language in the MS is generally well, but some phrases in Introduction and Discussion are too heavy constructed and long.
Author Response
Dear Reviewer,
Thank you very much for your time and all the comments and corrections to our manuscript. We agree with all the remarks and critical comments. We have taken all these into consideration and have incorporated the corrected text into the new version of the manuscript.
Once again, thank you very much.
Sincerely,
Maciej Skoracki